# Therapeutic Application of Extracellular Vesicles Derived from Mesenchymal Stem Cells in Domestic Animals

**DOI:** 10.3390/ani14152147

**Published:** 2024-07-24

**Authors:** Aliai Lanci, Eleonora Iacono, Barbara Merlo

**Affiliations:** 1Department of Veterinary Medical Sciences, University of Bologna, Via Tolara di Sora 50, Ozzano dell’Emilia, 40064 Bologna, Italy; eleonora.iacono2@unibo.it (E.I.); barbara.merlo@unibo.it (B.M.); 2Health Science and Technologies Interdepartmental Center for Industrial Research (HST-ICIR), University of Bologna, 40100 Bologna, Italy

**Keywords:** extracellular vesicles, exosomes, conditioned medium, microvesicles, mesenchymal stem cells, clinical application, fetal adnexa, adult tissues

## Abstract

**Simple Summary:**

Extracellular vesicles (EVs) secreted by mesenchymal stem cells (MSCs) have recently been described in human and veterinary medicine and great interest is directed toward them for their therapeutic potential. EVs are vesicles produced and secreted by Mesenchymal Stem Cells (MSCs) that possess the same anti-inflammatory and regenerative properties as the cells themselves but seem safer for clinical applications because of lower immunogenicity and lower size. MSCs that produce EVs can be derived from adult tissues such as bone marrow or adipose tissue or from fetal adnexa such as amniotic membrane and Wharton’s jelly. This manuscript gives a summary of the current description of composition, characteristics, sources, and function of EVs, focusing especially on their therapeutic applications in veterinary medicine to date. Knowledge of these features and the steps taken to date will help researchers and veterinary practitioners discover new approaches to regenerative medicine that are ever closer to being achievable.

**Abstract:**

Recently, the therapeutic potential of extracellular vesicles (EVs) derived from mesenchymal stem cells (MSCs) has been extensively studied in both human and veterinary medicine. EVs are nano-sized particles containing biological components commonly found in other biological materials. For that reason, EV isolation and characterization are critical to draw precise conclusions during their investigation. Research on EVs within veterinary medicine is still considered in its early phases, yet numerous papers were published in recent years. The conventional adult tissues for deriving MSCs include adipose tissue and bone marrow. Nonetheless, alternative sources such as synovial fluid, endometrium, gingiva, and milk have also been intermittently used. Fetal adnexa are amniotic membrane/fluid, umbilical cord and Wharton’s jelly. Cells derived from fetal adnexa exhibit an intermediate state between embryonic and adult cells, demonstrating higher proliferative and differentiative potential and longer telomeres compared to cells from adult tissues. Summarized here are the principal and recent preclinical and clinical studies performed in domestic animals such as horse, cattle, dog and cat. To minimize the use of antibiotics and address the serious issue of antibiotic resistance as a public health concern, they will undoubtedly also be utilized in the future to treat infections in domestic animals. A number of concerns, including large-scale production with standardization of EV separation and characterization techniques, must be resolved for clinical application.

## 1. Introduction

The study and application of mesenchymal stem cells (MSCs) has exponentially increased over the past decade since their versatility, safety and great potential exhibit a strong attraction for researchers [1]. In particular, MSCs are studied extensively for their application in regenerative medicine and tissue engineering [2] and they can be isolated from adult or fetal tissues [3,4]. In fact, there are a great number of in vitro and clinical studies on humans, and laboratory studies on small and large animals. The key to MSCs therapeutic potential is their ability to migrate to sites of tissue injury or inflammation, perceive hypoxia and tissue damage [5], stimulate endogenous repair of injured tissues [6], and modulate immune responses [7,8,9,10]. MSCs abilities are due to the fact they release many factors (composing the secretoma) into the environment (in vivo) or into the culture medium (in vitro) forming, in the latter case, the conditioned medium (CM). The secretoma is composed of small soluble factors (chemokines, cytokines, and hormones) and extracellular vesicles (EVs) that provide a vehicle for the transfer of lipids, proteins, and nuclear acids from one cell to another [11,12].

EVs are secreted from many cell types and differ with respect to their origin within the cell, size, and contents [13,14,15]. Since they are not only secreted by MSCs, the biological significance of EVs has for many years been largely overlooked, with them regarded like apoptotic bodies, as merely cellular fragments or debris [13,14,15]. By the most recent definitions [16,17] there are two main classes of EVs: exosomes, derived from budding of endosomal membranes and ranging in size from 40 to 100 nm, and microvesicles (MVs), which originate from outward budding and fission of the plasma membrane and which range from 50 nm to 1 μm in size. The diversity of proteins, lipids, and nucleic acids contained in EVs depends on their cell of origin and may be influenced by physiological stress or other conditions [14,18,19,20]. EVs have an important role in intercellular communication and are able to modify the activity of target cells through interactions with surface receptors and the transfer of proteins, mRNAs, and miRNAs [14]. EVs are responsible for the paracrine action of MSCs, exerting an effective mediating role that directly activates target cells, transferring information to damaged cells or stimulating adjacent cells to secrete other factors [16,21]. In particular, first, EVs may stimulate target cells directly by surface-expressed ligands, acting as signaling complexes [22]. Second, EVs may transfer surface receptors from one cell to another, deliver proteins, mRNA, bioactive lipids and even whole organelles (e.g., mitochondria) into target cells [22]. As an editorial noted, this fascinating EVs-mediated cell–cell communication system developed very early in evolution and served as a model for further development of intercellular interaction mechanisms involving soluble bioactive mediators and precision ligand–receptor interactions [22]. The earliest descriptions of EVs were performed in the early 1980s and EVs were initially thought to represent a mechanism for the elimination of proteins and other undesirable molecules [13,23]. Since the exact route that MSC-EVs follow from donor cells to recipient cells is still unknown, previous research has shown several important features [24]. EVs’ surface molecules have the ability to both direct them to recipient cells and prevent the inside components from deteriorating [24,25,26]. EVs can enter cells by a variety of routes once they are connected to a target cell, such as receptor–ligand contact, internalization through phagocytosis and/or endocytosis, or direct fusion with the plasma membrane [25,26]. Nevertheless, EVs have the capacity to interact with recipient cells by delivering a particular cargo directly into the cytoplasm [24,25,26].

The great attraction for the therapeutic use of EVs is derived from the reduced risks associated with MSCs grafting, from possible immune reactions against MSCs, and especially from the opportunity to load or modify the content of bioactive factors to address specific therapeutic needs [15]. Indeed, MSC-EVs have the same therapeutic potential as MSCs, which includes stimulating angiogenesis, encouraging cell migration and proliferation, inhibiting apoptosis, and reducing inflammation [24]. Furthermore, the quick development of EV-based treatment has been spurred by recent advances in nanotechnology. Because MSC-EVs are naturally biocompatible, biodegradable, and non-immunogenic, they are a promising class of nanomaterials for drug delivery applications [27].

These properties make MSC-EVs a viable method for EV-based therapies and they have been discovered to be relevant in a lot of fields. In human medicine, the main areas of application of EVs in preclinical studies conducted in laboratory animals have been cardiovascular disease [28,29,30], kidney [31,32,33], liver [34,35], lung injury and diseases [36,37], wound healing [38,39,40], tumor growth [41,42,43] ophthalmology [44,45], immune-related diseases [46,47,48,49,50], neurological diseases [51,52,53] and musculoskeletal system [54,55,56].

The present review will provide an overview of clinical–therapeutic applications of MSCs-EVs in veterinary medicine. First of all, the characterization of EVs derived from adult tissues and fetal adnexa is presented. Preclinical research carried out in sheep, pigs, and rodents will be mentioned in addition to the therapeutic applicability in vitro and in vivo for the main domestic animals. Furthermore, the future perspectives of the clinical application will be discussed to highlighting the importance of the challenges that need to be addressed and solved to make routine MSCs-EV therapy possible in domestica animals.

To conduct the present study, relevant information was sourced from databases such as Medline and Science Direct, Google Scholar, Pubmed, Scopus and CAB Abstracts. Multiple search queries were employed to find all related articles. Full-text reports in the form of reviews or research articles written in English language were included, while conference or congress contributions were excluded.

## 2. Characterization and Sources

EVs are nano-sized particles containing biological components commonly found in other biological materials [57]. For that reason, EV isolation and characterization are critical to draw precise conclusions during their investigation. In this context, the International Society for Extracellular Vesicles (ISEV) identified the minimal experimental requirements for the definition of EVs and their functions (MISEV 2014) [57]. Initially, they proposed this thorough documentation of the origin of materials and methods used for isolation, emphasizing their influence on the ability to replicate findings. Secondly, they advocated for a comprehensive assessment of the protein composition, including quantifying proteins anticipated to be concentrated in extracellular EVs. Moreover, they insisted on employing two complementary methods for characterizing individual particles. Lastly, the functional analysis should encompass appropriate control samples.

The initial suggestions underwent revision in 2018 [17]. MISEV2018 is presently regarded as the fundamental set of information to be disclosed when presenting findings involving EVs. The guidelines were enhanced by urging the provision of more detailed data concerning sample collection, as well as cell-culture specifics for EV gathering and storage. The reporting of isolation and concentration methods was emphasized, and no specific purification protocol was recommended. The catalog of methods for EV characterization was extensively updated to include recent advancements in single-particle techniques. Quantification of fundamental components like lipids, proteins, and RNA, along with their ratios (e.g., particle-to-protein ratio), was highlighted as crucial for assessing EV enrichment [17]. It was suggested that dose–response studies should be performed using EV preparations normalized to particle count or another approach of quantifying the biological cargo of the EVs, and to use multiple control samples, including distinct fractions of the enrichment protocol, in light of the significantly expanded list of specific protein markers [17]. Although EV research in veterinary medicine is still in its early phases, several studies have been published recently. It is strongly advised that specialized EV scientists and emerging EV-focused veterinary researchers work closely together to minimize biases and technical issues, provide optimal outcomes, and maximize the potential of group EV research initiatives [58].

### 2.1. Adult Tissues

The conventional adult tissues for deriving MSCs include adipose tissue and bone marrow. Nonetheless, alternative sources such as synovial fluid [59], endometrium [60], gingiva [61], and milk [62] have also been intermittently used. MSC-derived EVs have undergone purification and investigation across various domestic animal species including pig [63,64,65,66,67,68,69,70,71,72,73,74,75,76,77,78,79,80,81,82,83,84,85,86,87,88,89,90], horse [59,60,91,92,93,94,95,96,97,98,99,100,101,102,103,104], dog [61,105,106,107,108,109,110,111,112,113,114,115,116,117], cat [118,119,120,121,122], cow [62], and sheep [123]. While not all studies have comprehensively characterized EVs [58], most have involved isolation, purification, and characterization procedures focusing on size, morphology, and protein composition. Some studies have delved into nucleic acid content, whereas lipid quantification has been performed in only two studies focusing on Lyosecretome [98,112]. Tetraspanins (CD9, CD63, CD81) are commonly employed as identifying markers for EVs, although other biomarkers such as CD29, CD73, CD40, TSG101, β-catenin, and β1-integrins have also been utilized. The collective characterization of MSCs-EVs sourced from adult tissues in domestic animals is illustrated in Figure 1.

### 2.2. Fetal Adnexa

Cells derived from fetal adnexa exhibit an intermediate state between embryonic and adult cells [124,125,126,127], demonstrating higher proliferative and differentiative potential and longer telomeres compared to cells from adult tissues [128,129,130]. These characteristics are related to the early embryological origin of MSCs derived from fetal adnexa. Following blastocyst implantation, the inner cell mass undergoes morphological changes, leading to the formation of the bilaminar embryonic disc, which consists of the epiblast and the hypoblast. Some cells from the hypoblast migrate along the outer edges of the extraembryonic reticulum to form a connective tissue known as the extraembryonic mesoderm, which surrounds the yolk sac and amniotic cavity, later forming the amniotic mesoderm and chorionic mesoderm [131]. These cells, derived from extraembryonic mesoderm, are fundamental for maintaining feto–maternal tolerance during pregnancy, having important immunomodulatory characteristics and low immunogenicity [131,132]. Despite these important attributes of MSCs from fetal adnexa, MSCs-EVs have been purified and investigated only in equine and canine species [3,133,134,135,136,137,138,139,140,141,142].

In equine species, only Iacono et al. [3] noticed the presence of complex extracellular vesicles measuring 500 nM–1 μM, observed in MSCs from Wharton’s jelly (WJ) using TEM. In contrast, the research group of Lange-Consiglio reported in various studies [133,134,135] that amniotic membrane (AM)-MSCs produce EVs ranging in size from 100 nm to 1000 nm, with a predominance of vesicles between 100 and 200 nm, which they considered as shedding vesicles. In 2018, Lange-Consiglio et al. [135] performed miRNA sequencing of EVs derived from AM-MSCs for the first time, finding that EVs contain a lower percentage of miRNAs than AM-MSCs and that several miRNAs are enriched hundreds or thousands of times in EVs, while others remain at the same level as in AM-MSCs. Moreover, the authors reported that many of the miRNAs enriched in EVs regulate the inflammatory response, such as the overexpression of miR-146, which decreases the expression of the inflammatory cytokine IL-6 in lipopolysaccharide (LPS)-stimulated macrophage cells, and MiR-223, which negatively regulates the expression of many inflammatory genes in macrophage cells. The same research group supported these observations using AM-MSCs-EVs, in an in vitro model, to counteract the stress induced by LPS in endometrial, lung, and tendon cells [133,134,136]. In all cases, the authors observed the incorporation of EVs within cells and the downregulation of tumor necrosis factor-alpha (TNF-α), interleukin 6 (IL-6), IL-1β, matrix metalloproteinase-1 (MMP-1), and MMP-13 genes. Most recently, the same research group evaluated the surface glycosylation pattern of AM-MSCs-EVs released in conditioned medium, using a microarray procedure [137]. The signal intensity detected by microarray scanner indicated a high simultaneous presence of Galβ1,3GalNAc, α2,3 sialic acid, and high-mannose N-linked glycans, which the authors suggest may constitute markers of AM-MSCs-EVs in equine species.

Canine MSCs are of interest for both veterinary and comparative models of disease; however, in this species, as well as in humans, there are difficulties in acquiring adult tissues and there are ethical implications. For these reasons, in recent years, researchers’ attention has been directed towards identifying innovative sources of MSCs and EVs from easily accessible materials. In 2019, Crain et al. [138] reported that canine WJ-MSCs produce EVs of 125 nm in diameter. In this preliminary study on the mechanism of immune EVs modulation in canines, the authors observed that EVs inhibited CD4 T cell proliferation in a dose-dependent manner, hypothesizing a mechanism regulated by a TGF-βRI antagonist, neutralizing antibodies to TGF-β, or the A2A adenosine receptor blockade. Most recently, Wright et al. [139] characterized EVs derived from umbilical cord (UC) cells in canines, following MISEV guidelines. Canine UC-MSCS-EVs were found to be within the size range of exosomes (50–150 nm) with a median protein concentration of 3 g/mL. Isolated EVs were positively stained for CD9, CD63, CD81, ALIX protein, and CD142 (TF: Tissue Factor), similar to canine UC-MSCs. These findings indicate that EVs potentially share TF expression and potential pro-coagulant activity with MSCs of origin, which must be considered when using them as therapeutic agents. Regarding EVs derived from canine AM-MSCs, Karam et al. compared their morphological aspects at different culture passages, finding that, while the EV size did not differ between culture passages, their number decreased from passage 0 to passage 2 of in vitro culture [140]. Based on these observations, the authors suggest that cell–cell communication is greater in the early phase, making this the optimal phase for clinical EV application. These results were confirmed by Scassiotti et al. [141]. In line with the findings reported in human amniotic fluid and membranes, Pastore et al. [142] isolated EVs of different sizes enriched for Alix (ALG-2 interacting protein X), CD81 (Tetraspanin-28), and TSG101 (Tumor susceptibility gene 101) after sequential centrifugations. Since CD59 seems to be involved in regulating the immunomodulation of feto–maternal interaction during pregnancy, the authors investigated the expression of this cluster by AM-MSCs-EVs in canines. All isolated EVs fractions expressed CD59, indicating that EVs derived from the amnion carry a complement inhibitor and play a crucial role in promoting immune tolerance to embryo–fetal antigens and reducing the risk of abortion in dogs.

## 3. Clinical Application of EVs

The therapeutic potential of MSCs-EVs has been increasingly studied over the past decade.

### 3.1. Preclinical Studies

In pigs and sheep there has been less research than that on rodents, but it is all quite recent. Application areas in sheep include the musculoskeletal system, neurological system, and sepsis/pneumonia, while in pigs the respiratory and circulatory systems and renal/liver injury are currently being utilized. The main clinical studies conducted on these species are summarized in Table 1.

Only recently, some studies have characterized MSCs-EVs in the equine and canine species [101,141]. More recently, domestic animals, such as horse, dog, and cat have also been used in preclinical and clinical studies. Table 2 summarizes the main preclinical studies carried out in domestic animals. Regarding preclinical studies, EVs derived from adult and fetal tissues have been applied to cultures of tenocytes, chondrocytes, fibroblasts, and endometrial cells, during in vitro embryo production, and to alveolar macrophages, microglial cells and mesenchymal stem cells.

### 3.2. Clinical Studies/Applications

Table 3 summarizes the main clinical studies involving the in vivo application of EV-MSCs in veterinary medicine.

#### 3.2.1. Orthopedic Field

Two studies have been conducted in the orthopedic field: one study involved one horse with ligament injury [96], and three dogs with spontaneous osteoarthritis [98]. In the ligament injury, the injection of 25 μg/mL EVs increased angiogenesis and elasticity in the ligament injury and also promoted lesion filling without adverse reaction [96]. The study conducted in the dog used a product called Lyosecretome (freeze-dried secretome), which has previously created by the same research group using MSCs obtained from human adipose tissue [150]. In order to concentrate and purify the MSC-derived secretome, this procedure included an ultrafiltration phase. After that, the secretome was freeze-dried to provide a powdered dosage form with improved long-term stability [98]. Dogs suffering osteoarthritis received 20 mg, or 2 × 10^6^ cell equivalents, resuspended in hyaluronic acid for application [98]. Allogeneic Lyosecretome injected intra-articularly is safe and does not cause a clinically relevant systemic or local adverse response [98].

#### 3.2.2. Reproductive Field

In the reproductive field, two studies have been conducted on mares with chronic endometritis [103,149]. A case report was conducted on an 11-year-old Friesian mare with a history of failed pregnancies despite numerous insemination attempts [150]. Two treatments with 20 billion EVs diluted in 50 mL of NaCl 0.9% were performed, followed by an uterine biopsy. The success of the intrauterine administration of EVs is demonstrated by an improvement in the classification of endometritis and in a successful artificial insemination with implantation of an embryo, as detected at day 14, and with a pregnancy that is still ongoing [149]. Day-8 equine embryos are thought to release EVs that transfer early pregnancy factors including HSP10 and miRNA, hence modulating the function of the oviductal epithelium [151]. The endometrial epithelium also secretes EVs, which target cellular pathways important for embryo implantation [152]. Pro-inflammatory cytokines, growth factors, and chemokines operate on fibroblasts and other cells in the physiopathological mechanisms of equine endometriosis, as well as in other inflamed tissues, influencing extra-cellular matrix deposition and tissue fibrosis [153].

In the second study of chronic endometritis, 14 mares were included and were divided into control and endometritis groups [103]. During the first and second ovulation, EVs were injected twice, separated by 21 days; 400 μg/mL of EVs was added to the sterile carboxymethylcellulose (CMC) at a concentration of 22 mg/mL. Each mL of the gel contained 200 μg EVs. The CMC gel with MSCs-EVs is preserved by lyophilization, and lyophilized EV solution was administered intrauterinely. Doppler and hormonal analysis were performed in addition to uterine biopsy [103]. After the second EV treatment, the histological evaluation revealed the regression of fibrous tissue and restoration of healthy endometrial glands with normal epithelium. In all treated mares, on the ninth day after insemination, an embryonic vesicle, and thus a pregnancy, was identified. The activity of EVs and the miRNAs contained in them triggered tissue regeneration, resulting in a restoration to the original histological features and thus, normal endometrial function. This probably restored the conditions needed for the appropriate implantation and development of maternal–embryonic paracrine communication [103].

The third study concerns the persistent post-breeding-induced endometritis, which is considered a major cause of subfertility in mares. Authors identified an optimal concentration of 400 × 10^6^ EVs with 10 × 10^6^ spermatozoa/mL: at this concentration, sperm mobility parameters were not negatively affected [149]. Semen alone or semen enhanced with EVs was used for insemination of sixteen susceptible mares. The supplementation resulted in a decrease in intrauterine fluid accumulation and polymorphonuclear neutrophil infiltration, along with a noteworthy decrease in intrauterine TNF-α and IL-6 and an increase in anti-inflammatory IL-10 in mares in the EV group, indicating effective regulation of the post-insemination inflammatory response [149].

Comparing these studies is challenging because of the limited sample size, and the substantial difference between the two considered diseases with a distinct etiology, as well as the different sources of EVs used (AM and BM). Despite this, the results are encouraging, although the different authors also measured and evaluated different molecules in addition to the common result represented by the positive diagnosis of pregnancy. A large clinical trial should be conducted on the problem-mares with persistent post-breeding-induced endometritis and chronic endometritis by standardizing the amount of EVs and the protocol of administration. Results should be supported by a histologic examination, in addition to a positive pregnancy diagnosis.

In cattle, the embryo culture was supplemented with or without 100 × 10^6^ EVs/mL in Holstein Friesian cows, and this seemed to partially modify the expression of certain miRNAs involved in successful embryo implantation and prevent the detrimental effects of in vitro culture [146].

#### 3.2.3. Skin-Wound Field

Regarding skin wounds, El-Tookhy et al. [105] used a dog full-thickness skin defect model. The wounds were induced using a dermal punch and the wounds were 2.5 cm apart. The results showed that MSCs-EVs significantly accelerated and increased cutaneous wound healing, collagen synthesis, and vascularization at wound sites, and showed faster wound closure. Additionally, it was determined that the application of EVs sped up the maturity of freshly created capillaries at wound sites, in addition to encouraging the formation of new ones [105] The amount of EV injected in this study equals the amount produced by a 2 × 10^6^ MSCs/1 mL/wound [105].

One of the more complex biological processes that may be observed is wound healing, which requires coordinated action between cells, growth factors, and extracellular matrix proteins. Conducting well-designed studies is crucial to compare the literature and obtain practical insights that might aid clinicians, especially considering the substantial amount of research that has been carried out on this topic in both human and veterinary medicine. Dogs provide ideal research models, and studies on wound healing have used them as translational models for both human and veterinary applications [154].

Induced-wound sizes used in this study were performed considering the critical size defect for dogs [105]. Future trials should consider different types of wounds, such as those that have arisen spontaneously and those that are more difficult to heal such as pressure sores, as performed in other studies in which MSCs have been applied [155,156,157].

#### 3.2.4. Urinary Tract

Artificial urinary occlusion was used to create the cat models of PR-AKI, which were subsequently treated with MSCs-EVs [122]. The infusion dose was equivalent to the amount of EV secreted by 10 million allogenic feline AT-MSCs in 24 h. Treatment with EVs was found to be effective in restoring plasma phosphorus, urea nitrogen, and creatinine. A routine blood examination revealed that the PR-AKI cats treated with EVs had faster return of their leukocytes to the normal physiological range than the control group. The plasma metabolome profile of PR-AKI cats treated with EVs was shown to be strikingly comparable to that of normal cats using ultra-high performance liquid chromatography analysis. Additionally, the examination of plasma demonstrated a strong correlation between the dynamic process of PR-AKI in cats and six metabolites found in plasma: carnitine, melibiose, d-glucosamine, cytidine, dihydroorotic acid, and stachyose [122]. The study demonstrated the efficacy of treatment with MSC-EVs and also discovered new PR-AKI indicators in addition to six metabolites, which may be potential targets for MSC-EVs treatment [122].

In the dog, Liu et al. 2023 [116] included 20 dogs as a model of renal ischemia-reperfusion injury, and the renal cortex of the left kidney was injected with EVs (180 μg/kg) in the experimental group. The EV treatment group showed reduced mitochondrial damage and a decrease in mitochondrial number as compared to the renal ischemia-reperfusion injury model group [116]. Renal ischemia-reperfusion injury resulted in severe histological abnormalities and significant increases in markers for renal function, inflammation and apoptosis, which were reduced by the infusion of MSCs-EVs [116].

#### 3.2.5. Mammary Gland

Regarding the application of the mammary gland in cattle, 48 animals were enrolled, of whom 32 had acute mastitis and 16 had chronic mastitis [147]. After milking, the treated cows received CM through intramammary application with 3 mL of CM alone or, in the control group, they received the antibiotic alone, intramammarily, chosen from an antibiogram test, for three consecutive days. The standard experimental volume of CM was set at 3 mL since, after lyophilization, CM was concentrated 4-fold, and a uniform treatment protocol was needed to compare this to the antibiotic treatment [147]. In vitro results showed that the addition of CM inhibited CFU bacteria and decreased bacterial growth; in cell culture infected with *S. aureus*, cells died in 12 h, while with the addition of CM 60–89% of cells remained viable. Regarding the in vivo application, there was no difference in the improvement of clinically affected quarters treated with CM compared to antibiotic treatment, but the rate of relapse was different. There was no statistically significant difference between the antibiotic group compared to the CM groups but, in the antibiotic treatment, the mean value of somatic cell count decreased, compared to the CM treatments. The study showed that treating mastitis with CM would reduce the need for antibiotics, minimizing antibiotic resistance and preventing the need for costly and inefficient treatments. The mammary gland appears to be trying to repair itself, and, in this situation, using CM, which is high in growth factors, could facilitate the regenerative process [147]. Furthermore, animals with chronic mastitis are frequently culled from the herd rather than receiving treatment. In the future, this research could help restore milk production even in cows who would otherwise be forced to leave the production cycle, in addition to significantly reducing the use of antibiotics. The successful recovery of glandular tissue and the prevention of antibiotic residues in milk could be two additional financial benefits of using MSCs-CM [147].

### 3.3. Future Perspectives and Challenges in Clinical Applications

Compared with MSC therapy, the use of MSCs-EVs offers more advantages. In fact, MSCs-EVs are highly stable, suitable for long-term storage, and can induce intercellular communication by directly transferring functional proteins and miRNAs [158]. Furthermore, allogeneic administration of EVs does not result in an immune response, and their ability to avoid the possible carcinogenicity of MSCs is a major benefit [158].

Treatment of infectious disorders is another clinical area where potential EV therapy is growing, according to recent studies. Infectious diseases that were thought to be under control are becoming more severe and new ones are emerging [159]. The effectiveness of antibiotic treatment of these diseases is threatened by antibiotic resistance. In fact, worldwide, one of the most significant problems endangering the health of humans and animals is antibiotic resistance (AMR) [160]. Alternative therapeutic approaches to treat infections are becoming increasingly essential, as a result of the failure of regularly used treatment approaches and an increase in the number of outbreaks of serious infectious diseases [159]. Nowadays, the majority of research on infectious diseases has been conducted on rodents, including studies on sepsis, lung infections, wound infections, urinary tract infections and intestinal infections. Regenerative tissue, direct antibacterial effects, and immunomodulation are the therapeutic mechanisms [159]. Given the importance of reducing antibiotic therapy in order to decrease antibiotic resistance for public health, it is certain that numerous studies on domestic animals will also be carried out in the coming years.

Despite the encouraging results of preclinical and clinical studies of the application of MSCs-EVs in domestic animals, there are several challenges to overcome in order to achieve routine clinical application, such as large-scale production and precise isolation methods, taking into account and reducing factors that affect EV quality and quantity, finding rapid and accurate characterization of EVs, the precise content of the EV cargo, and the safety profile.

### 3.4. Large-Scale Production, Isolation Methods and Factors That Affect EV Quality/Quantity

Although MSCs are relatively easy to expand using conventional tissue flasks and bioreactors, their growth in culture is finite and their biological properties may become altered with repeated passage. There is an urgent need for development of methods for reliable expansion of MSCs to mass-produce EVs for clinical use. In fact, current methods of expansion of MSCs are labor-intensive and involve several procedures. In order to facilitate large-scale MSC-EV production, new batches of MSCs will have to be periodically derived, with significant impact on the costs [161]. To overcome this limitation different methods could be used, such as immortalization by natural selection or by genetic modification or clonal isolation [162,163]. A recent study in human medicine showed that hUC-MSC culture in scalable three-dimensional cultures resulted in a twenty-fold greater yield of EV than two-dimensional cultures [164]. In human medicine, several research groups have demonstrated that EVs isolated from MSCs culture by ultrafiltration followed by size-exclusion chromatography results in a higher yield while preserving EVs’ biophysical and functional properties [165,166,167,168].

The production of MSCs-EVs can be influenced by a number of factors, including cellular confluence, early vs. later cell passages, oxygen concentration, cytokines, and medium serum content [169]. A recent standard protocol for Good Manufacturing Practices (GMPs) offers a solution to produce MSCs and MSC-EVs on a large scale [170]. A similar procedure could be carried out with MSCs-EVs derived from horse and dog in veterinary medicine.

### 3.5. Safety Profile

Establishing a safety profile is another fundamental condition for clinical application. The greatest apprehension about the in vivo therapeutic use of cells is the differentiation of the transplanted MSCs and the potential of MSCs to suppress anti-tumor immune responses and to act as a progenitor for blood vessels, which potentially promote tumor growth and metastasis [171].

## 4. Conclusions

In the present review, we have summarized what EVs are, and that they can be readily isolated from MSCs derived from adult and fetal tissues. MSCs-EVs are known to have therapeutic benefits in different animal disease models and have theoretical advantages over intact MSCs as a therapeutic product, and in the future may be preferred over whole cells in the regenerative medicine field. For clinical application, several issues need to be addressed, such as large-scale production with standardization of EV isolation and characterization protocols.

## Figures and Tables

**Figure 1 animals-14-02147-f001:**
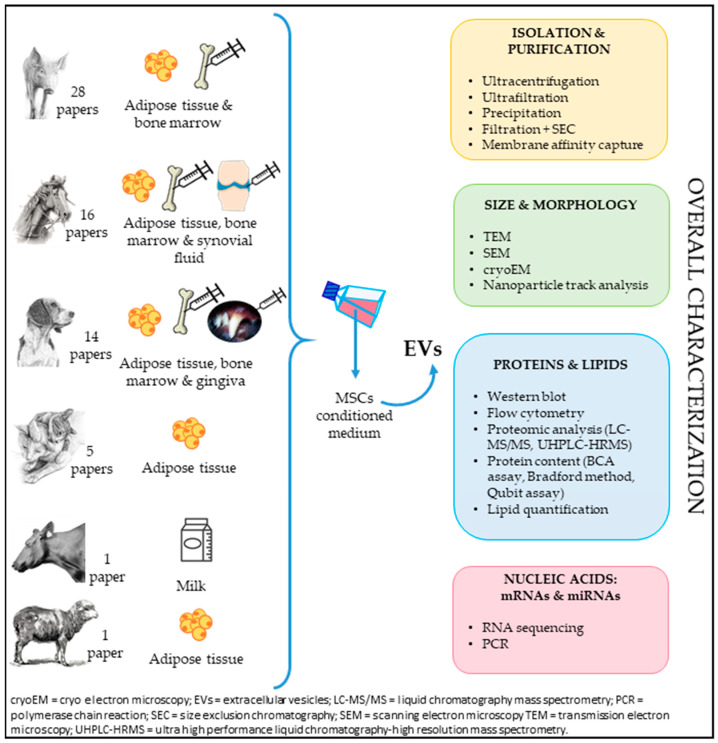
Source and overall characterization of MSCs-EVs in domestic animals.

**Table 1 animals-14-02147-t001:** Application of MSCs-derived EVs in pig and sheep. p = porcine; h = human; BM = bone marrow; AT = adipose tissue; UC = umbilical cord.

Animal	Source of EVs	District	Application	Mode of Administration	Reference
Pig	pBM-MSCs	Respiratory	Influenza virus	Intratracheal	[70]
	pAT-MSCs	Cardiovascular	Myocardial infarction	Intraoperative insertion of EVs combined with biocompatible cardiac scaffolds	[81]
	pBM-MSCs	Cardiovascular	Myocardial infarction	Intraoperative insertion of EV collagen patch	[90]
	pAT-MSCs	Urinary	Model of metabolic syndrome and renal artery stenosis in cardiovascular complications	Intrarenal injection	[67,68,75,77,78,79,80,82,83]
	pAT-MSCs	Vertebral column	Inducted spinal cord injury	Intrathecal injection	[89]
	pAT-MSCs	Hepatic	Inducted liver injury (hemi-hepatectomy and hepatic ischemia-reperfusion injury)	Intravenous	[87,88]
Sheep	hBM-MSCs	Neurological	Hypoxic ischemic encephalopathy (HIE)	Intravenous	[143]
	hUC-MSCs	Musculoskeletal	Ligament injury	Application onto a type 1 collagen sponge	[144]
	hBM-MSCs	Systemic	Pneumonia/sepsis	Intravenous	[145]

**Table 2 animals-14-02147-t002:** Preclinical studies in domestic animals with the use of EV/CM in culture. AM = amniotic membrane; BM = bone marrow; AT = adipose tissue; Syn Fluid = synovial fluid; WJ = Wharton’s jelly; e = equine; b = bovine; c = canine; f = feline.

Animal	Source of EVs	Application Culture	Effects	Reference
Horse	eAM-MSCs	Endometrial cells	reduced the apoptosis rate, increased cell proliferation values, downregulated pro-inflammatory gene expression, and decreased the secretion of pro-inflammatory cytokines	[134]
	eAM-MSCs	Tenocytes	induced a down-regulation of MMP1, MMP9, MMP13 and TNFα expression	[130]
	eBM-MSCs eAT-MSCs ad Syn Fluid	Chondrocytes	reduced inflammation	[59]
	eBM-MSCs	Chondrocytes	increased the articular chondrocyte collagen protein amounts, mRNA levels of Prg4, and enhanced the proliferation and migratory capacities of chondrocytes	[99]
	eBM-MSCs (autologous)	Chondrocytes	anti-inflammatory effects on gene expression following chondrocyte exposure to tumor necrosis factor α and Interleukin 1β	[97]
	eBM-MSCs	Chondrocytes	induced a greater increase in equine articular chondrocyte-neosynthesized hyaline-like matrix by modulating collagen levels, increasing PCNA, and decreasing Htra1 synthesis	[104]
	eAM-MSCs	Alveolar macrophages	Modulatory-effect release of TGF-alfa and β and possibly IL-6	[133]
Bovine	bAM-MSCs	Blastocysts	addition of EVs during in vitro embryo production seemed to influence the developmental capacity and implantation potential of the embryos and regulate the expression of specific miRNAs that regulated blastocyst development	[146]
	bAM-MSCs (CM)	Mammary epithelial cells	could attenuate bacterial growth, as evaluated by the number of CFUs. After 24 h of culture with *S. aureus*, 89.67% of mammary epithelial cells treated were still alive, whereas all cells cultured and not treated were dead	[147]
Dog	cWJ-MSCs	Fibroblasts	suppressed the proliferation of cell T CD4+ using TGF-β and adenosin	[138]
	cBM-MSCs	Murine microglia cells	decreased inflammation (decrease IL-1β)	[113]
	cAT-MSCs (Lyosecretoma)	Tenocytes, chondrocytes and AT-MSCs	induced proliferation of cells in dose-dependent manner and showed anti-elastase activity	[98]
	cAM-MSCs	Coculture with AM-MSCs and AT-MSCs	15–20% increased expansion rate	[141]
	cAT-MSCs	Semen during cryopreservation	initiated damaged-sperm repair (higher motility, live sperm percentage, membrane and acrosome integrity; higher expression of genes related to the repair of plasma membrane and chromatin material) and decreased reactive oxygen species production	[106]
Cat	fAT-MSCs and fibroblasts	Human THP-1 Macrophages	MSCs-EVs had lower levels of pro-inflammatory cytokines (IL-1β, TNF-α) and higher level of IL-10. MSCs-EVs played a crucial role in immune defense compared with EVs–fibroblasts	[118]

**Table 3 animals-14-02147-t003:** Application of EVs/CM (culture medium)-derived MSCs in veterinary medicine. e = equine; b = bovine; c = canine; f = feline; AT = adipose tissue; AM = amniotic membrane; BM = bone marrow; UC = umbilical cord.

Animal	Source of EVs	District	Application	Mode of Administration	Reference
Horse	eAT-MSCs	Musculoskeletal	Ligament injury	Ultrasound-guided injection at the injury site	[96]
	eAT-MSCs	Musculoskeletal	Osteoarthritis	Intra-articular injection	[101]
	eAM-MSCs	Reproductive	Chronic endometritis	Intrauterine	[148]
	eBM-MSCs	Reproductive	Chronic endometritis	Intrauterine	[103]
	eAM-MSCs (CM)	Reproductive	Persistent post-breeding-induced endometritis	Intrauterine	[149]
Cattle	bAM-MSCs	Reproductive	Blastocyst development	In vitro during embryo production	[146]
	bAM-MSCs (CM)	Mammary gland	Acute and chronic mastitis	Intramammarily	[147]
Dog	cBM-MSCs	Skin	Inducted skin wound	Subcutaneous injection	[105]
	cAT-MSCs (Lyosectetoma)	Musculoskeletal	Osteoarthritis	Intra-articular injection	[98]
	cAT-MSCs	Urinary	Renal ischemia-reperfusion injury	Renal cortex injection	[116]
Cat	fAT-MSCs	Urinary	Post-renal acute kidney injury (PR-AKI)	Intravenous	[122]

## Data Availability

No new data were created or analyzed in this study.

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
