# Peer review of "Therapeutic Application of Extracellular Vesicles Derived from Mesenchymal Stem Cells in Domestic Animals"

_animals, 2024, doi:10.3390/ani14152147_

Round 1

Reviewer 1 Report

Comments and Suggestions for Authors

The section on equine studies could be strengthened by including more critical analysis or discussion on the limitations of the studies reviewed, such as sample size variations or methodological inconsistencies across different trials.

The canine studies section effectively demonstrates the therapeutic benefits of MSC-EVs in wound healing. However, it lacks depth in discussing potential variations in outcomes based on different types of wounds or severity of injuries treated. Providing such insights would enhance the practical applicability of the findings.

In the mammary gland section you could consider expanding on how MSC-EVs specifically improve outcomes compared to standard treatments and discuss any potential drawbacks or challenges encountered.

Line 16: the word "man8uscript" must be corrected

Line 69:"many cells type" should be "many cell types."

Line 383:"take into account and reduce factors" should be "taking into account and reducing factors."

Reviewer 2 Report

Comments and Suggestions for Authors

Extracellular Vesicles (EVs) are a group of cell-derived membranous structures comprising various components, which involved in multiple physiological and pathological processes. Cell tunes EV biogenesis depending on its physiological state and releases EVs with particular lipid, protein and nucleic acid, which are being developed for use as biomarkers, modulators, also as cargo vehicles for the targeted delivery of compounds.

Mesenchymal stem cells (MSCs) are multipotent cells with many therapeutic applications. MSC-derived EVs have become a new direction for the treatment of diseases, which are being explored as a therapeutic alternative to MSCs since they may have similar therapeutic effects but are cell-free.

This manuscript not only summarizes the types and clinical applications of vesicles, but also discusses the production, application and safety of MSC-derived EVs. The authors provide a valuable overview of the potential application of MSC-derived EVs as a therapeutic in veterinary medicine.

There are few areas need attention:

 Given MSC-derived EVs as a therapeutic in treatment of diseases is an exciting and active field, the introduction seems to be a little basic and could have included a number of related studies.

Line 16: The word " man8uscript " is misspelled.

Figure 1: Please improve the clarity of the image.

Reviewer 3 Report

Comments and Suggestions for Authors

This manuscript has some good information, but appears to have been written as more of a journal article than a scientific review. Biased statements like exceptional and undoubtedly are used rather than neutral language.

The biggest concern that I have is that the focus is on EVs rather than the contents of EVs. The reason for the MISEV information is that there is variability in the contents and effects of EV based on their composition. A comprehensive review should tackle what in each EV is eliciting the clinical effects not that EVs have those effects on their own.  

Please include the criteria for paper selection and omission in this review including database searches and number of papers identified.

The manuscript’s organization is not intuitive. Please include/expand the description of the paper’s organization in the introduction.

For the literature reviewed, please include if each ISEV experimental requirements and disclosed the information stipulated by MISEV2018. Perhaps this can be incorporated in a table or explained as part of the paper inclusion criteria.

Figure 1 is misleading. It appears as those same EV characterization was performed for all EVs. If that is not the case, consider revising into a table or omitting.

Section 3 (starting on line 208) appears to address preclinical studies (207) this is confusing since the preclinical section starts on line 223 and the clinical section starts on line 237. Consider omitting 207-223 as this is at best redundant information.

Omit figure 2. This is basically a list of rodent applications and is superfluous.

What do you mean by “less advanced” in line 224.  The number of studies does not mean that the studies were not “advanced” please revise.

Table 1 Use of capital and formatting is not consistent. The effects associated with reference 142 should be “reduced”  rather that “reduces”. For 66 should say “Increased eAC...” For 71 Use an alternate to “improvement”. 143 should say “seemed…regulated”. 105 “Suppressed”, 80 “Decreased”, 65 “induced”, 108 “increased”, 73 cut 1st nice words (bias) start at “Initiated..”

Starting on line 258. The manuscript becomes more biased and starts using less objective language. All references to improved or improvements should be changed to more specific and less objectives terms like increased/decreased, etc.  

Revise the following sections to increase clarity. Line ending at line 45, 55, 62, 66, 93, 113, 420 and paragraph ending line 115, 120, 262, 298, 408,

Provide a references for the following statements or revise. Line ending on line 58, 65, 68, 81, 112, 151, 208, 362, 385

Cut line 239-249 (redundant).

Cut line 249-57 (biased).

Cut Line 366-367 (superfluous)

Cut 397-400 (needs to support relevance)

Cut line 415-416 (biased)

Cut line 421-423 (biased)

Remove “exceptional” from line 419 (bias)

A number of errors and inconsistencies exist throughout and reflect on the quality of the manuscript.

Some sections (including tables) are in present tense and others are in past tense.

Please addressing spelling and editing errors line 16, 43, 51, 160, 178, 196, 238, 239, 259

Please standardize use of italics in words like in-vivo and in-vitro

Please reformat references. These are inconsistent and do not all meet journal standards. Please watch spacing, use of capitals and journal name abbreviations.

Comments on the Quality of English Language

A number of errors and inconsistencies exist throughout and reflect on the quality of the manuscript.

Some sections (including tables) are in present tense and others are in past tense.

Please addressing spelling and editing errors line 16, 43, 51, 160, 178, 196, 238, 239, 259

Please standardize use of italics in words like in-vivo and in-vitro

Please reformat reference. These are inconsistent and do not all meet journal standards. Please watch spacing, use of capitals and journal name abbreviations.

Author Response

Please the attachment 
